# Supplementation with *Saccharomyces boulardii* Increases the Maximal Oxygen Consumption and Maximal Aerobic Speed Attained by Rats Subjected to an Incremental-Speed Exercise

**DOI:** 10.3390/nu11102352

**Published:** 2019-10-02

**Authors:** Anne Danieli Nascimento Soares, Samuel Penna Wanner, Elissa Stefane Silva Morais, Alexandre Sérvulo Ribeiro Hudson, Flaviano Santos Martins, Valbert Nascimento Cardoso

**Affiliations:** 1Department of Clinical and Toxicological Analysis, Faculty of Pharmacy, Universidade Federal de Minas Gerais, Belo Horizonte, MG, 31270-901, Brazil; annedanieli@gmail.com (A.D.N.S.); elissamorais_@hotmail.com (E.S.S.M.); valbertcardoso@yahoo.com.br (V.N.C.); 2Instituto Federal de Educação, Ciência e Tecnologia do Sudeste de Minas Gerais, Barbacena, MG, 36205-018, Brazil; 3Exercise Physiology Laboratory, School of Physical Education, Physiotherapy and Occupational Therapy, Universidade Federal de Minas Gerais, Belo Horizonte, MG, 31270-901, Brazil; alexandre.servulo@yahoo.com.br; 4Department of Microbiology, Institute of Biological Sciences, Universidade Federal de Minas Gerais, Belo Horizonte, MG, 31270-901, Brazil; flaviano@icb.ufmg.br

**Keywords:** aerobic capacity, metabolic rate, microbiota, physical performance, probiotics, yeast

## Abstract

Benefits to the host metabolism resulting from *Saccharomyces boulardii* (Sb) supplementation have been described; however, no study has investigated the effects of this supplementation on aerobic metabolism and performance during physical exercise. Thus, in the present study, we addressed the effects of Sb supplementation on the rate of oxygen consumption (VO_2_), mechanical efficiency (external work divided by VO_2_), and aerobic performance of rats subjected to fatiguing, incremental-speed exercise. Twenty-six male Wistar rats were randomly divided into two groups: (1) non-supplemented, in which rats received 0.1 mL of a saline solution, and (2) Sb-supplemented, in which rats received 0.1 mL of a suspension containing 8.0 log_10_ colony-forming units. The rats received the treatments by gavage for 10 consecutive days; they were then subjected to fatiguing treadmill running. Sb supplementation did not change the VO_2_ values or mechanical efficiency during submaximal exercise intensities. In contrast, at fatigue, VO_2MAX_ was increased by 12.7% in supplemented rats compared with controls (*p* = 0.01). Moreover, Sb improved aerobic performance, as evidenced by a 12.4% increase in maximal running speed attained by the supplemented rats (*p* < 0.05). We conclude that Sb supplementation for 10 days increases VO_2MAX_ and aerobic performance in rats.

## 1. Introduction

Probiotics are defined as live microorganisms that, when administered in adequate amounts, confer a health benefit to the host [1,2]. In this context, *Saccharomyces boulardii* (Sb) is a nonpathogenic yeast marketed worldwide due to its probiotic properties [3]. Sb regulates intestinal microbial homeostasis and, therefore, modulates both local and systemic immune responses [3].

Benefits to the host metabolism resulting from Sb supplementation have also been described. For instance, Sb supplementation was shown to attenuate the increase in serum concentrations of triglycerides and total cholesterol induced by a cholesterol-enriched diet in hamsters [4] and to reduce the body weight, fat mass, hepatic steatosis, and inflammatory tone in obese/type 2 diabetic mice [5]. Moreover, Sb supplementation increases the fecal concentrations of short-chain fatty acids (SCFAs) in patients undergoing enteral nutrition [6], and these gut-derived SCFAs play important roles as substrates for glucose, cholesterol, and lipid metabolism [7]. However, whether these benefits promoted by Sb also result from changes in the aerobic metabolic rate remains to be investigated.

Physical exercise is a condition that places a burden on body metabolism due to the augmented energy demand of contracting muscles. The ability to perform prolonged exercise (i.e., endurance or aerobic capacity) is paramount in many sports and occupations. Recent literature has provided evidence that supplementation with a multi-strain probiotic for four weeks can improve human aerobic performance in the heat, as indicated by an increased running time to fatigue [8]. Of note, these findings did not confirm the findings of a previous study by Cox et al. [9], where no differences were reported in the treadmill performance of trained distance runners following supplementation; these two studies treated the individuals with different bacterial genera. As far as we know, no study has investigated the effects of supplementation with a yeast that acts as a probiotic (i.e., Sb) on aerobic performance.

The mechanisms underlying improved aerobic performance involve adaptations in multiple physiological systems, including skeletal muscles. For example, muscle adaptations are characterized by greater mitochondrial volume and increased expression of mitochondrial enzymes involved in β-oxidation, the tricarboxylic acid cycle, and the electron transport system [10]. The ability to increase the rate of chemical to mechanical energy conversion via aerobic metabolism, to spend less energy when exercising (i.e., improved running economy), and to depend less on the glycolytic pathway for energy conversion (i.e., a rightward shift in the lactate threshold) are some of the factors that are dependent on muscle functioning and determine the capacity to exercise for prolonged periods [11]. Previous studies showed that aerobically trained rats present a lower energy cost of running [12,13] and greater maximal rate of oxygen consumption (VO_2MAX_) [13]. These adaptations may underlie the performance-enhancing effects of probiotics, which modulate body metabolism as described earlier.

Thus, the present study investigated the effects of Sb supplementation on the rate of oxygen consumption (VO_2_) of rats kept under resting conditions or subjected to fatiguing, incremental-speed treadmill running. We also evaluated the running economy, mechanical efficiency, and maximal speed attained by rats subjected to exercise. We expected greater aerobic performance in rats treated with Sb, which would be associated with greater VO_2MAX_, running economy, and mechanical efficiency.

## 2. Materials and Methods

### 2.1. Microorganisms

Viable *Saccharomyces boulardii* cells were used after isolation on Sabouraud dextrose agar (Difco) from a lyophilized commercial preparation (Floratil, Merck S.A., Kenilworth, NJ, USA). The yeast was grown overnight at 37 °C with constant shaking (180 rpm) in YPD (yeast extract 1%, peptone 1%, and dextrose 2%) broth. The culture was then concentrated to obtain 10^9^ (9.0 log_10_) colony-forming units (CFU)·mL^−1^; for treatment, rats received 0.1 mL of this suspension by gavage, which means that each animal was treated with 10^8^ CFU (or approximately 3 × 10^8^ CFU·kg^−1^·day^−1^).

### 2.2. Animals

Twenty-six male Wistar rats (11 weeks old) weighing 250 to 350 g were provided by the Animal Care Center at the Faculty of Pharmacy of the Universidade Federal de Minas Gerais. The rats were housed in individual cages under controlled light (05:00–19:00 h) and temperature (24.0 ± 2.0 °C) conditions with water and commercial chow (Nuvilab CR-1) provided ad libitum. The experiments were approved by the local Ethics Commission on Animal Use (protocol number: 34/2014) and complied with the policies determined by the National Council for the Control of Animal Experimentation (CONCEA/Brazil).

### 2.3. Experimental Design

The rats were randomly allocated into two groups, with 13 rats in each group: (1) non-supplemented (NS), in which rats received 0.1 mL of a saline solution, and (2) supplemented (Sb), in which rats received a suspension containing Sb. The rats received the treatments by gavage and had food intake and body mass recorded once per day for 10 consecutive days. Since our objective was to test the effectiveness of a preventive intervention aimed at protecting an organism during subsequent exposure to stressful stimuli, a 10-day supplementation period was chosen. In this sense, we demonstrated that mice supplemented with Sb for 10 days had decreased disease and death caused by an experimental model of typhoid fever [14].

During 5 consecutive days of the supplementation period (from the 4th until the 9th day), all of the rats were familiarized with running on a treadmill. On the 10th day of the supplementation, each rat was subjected to an incremental-speed exercise. All experimental trials were performed during the light phase of the day. Following the trial, each rat was euthanized with a lethal dose of anesthetic (a cocktail containing 240 mg·kg^−1^ of ketamine and 31.5 mg·kg^−1^ of xylazine) administered via intraperitoneal injection.

### 2.4. Familiarization with the Running Exercise on a Treadmill

The familiarization protocol consisted of running on a treadmill designed for small rodents (Panlab, Harvard Apparatus, Cornellà, Spain) over 5 consecutive days. Rats were encouraged to run by light electrical stimulation (0.2 mA) provided by a grid located at the rear end of the treadmill belt. Each familiarization session initially consisted of 5 min during which the rat could move freely on the treadmill belt; the treadmill was then turned on, and the speed was gradually increased up to 15 m·min^−1^, at which point the speed was held constant for 5 min [15]. At the last familiarization session, the rats could run with minimal exposure to electrical stimulation. The incline was always set at 5% during all familiarization sessions and experimental trials.

### 2.5. Experimental Trials

On the day of the experimental trials, the rats were weighed, administered the assigned treatment by gavage, and then transferred from their home cages to the motor-driven treadmill. The chamber that contained the treadmill belt was sealed. VO_2_ was measured by open-circuit indirect calorimetry. For this measurement, the motor-driven treadmill chamber was coupled to a gas analyzer (Panlab, Harvard Apparatus), and the air flow inside the chamber was maintained at 2.0 L·min^−1^. The gas sensors were calibrated with primary gas standards containing known concentrations of O_2_ and CO_2_. The air leaving the chamber was automatically sampled and passed through the gas analyzer to determine the O_2_ and CO_2_ content of the air. The VO_2_ was measured every second, and 1-min average values were calculated with the help of Metabolism 2.2v software.

The rats were allowed to rest for 90 min and were then subjected to an incremental-speed exercise. The exercise started at a speed of 10 m·min^−1^ with increments of 1 m·min^−1^ every 3 min until the animals were fatigued [16]. Fatigue was determined as the moment when the rat could not maintain the pace with the treadmill, subjecting itself to the light electrical stimuli for 10 s [15]. VO_2_ was measured continuously during resting and exercise sessions. The ambient temperature was controlled at 24 °C. It is worth mentioning that physical performance measured during this exercise (i.e., workload) has been shown to be positively correlated with the VO_2MAX_ of untrained rats [12].

Three different indices were used to determine aerobic performance: Time to fatigue, the maximal speed attained, and the external work performed by the rats. The maximal aerobic speed (S_MAX_) attained during the incremental exercises was calculated according to the equation described by Kunstetter et al. [16]: S_MAX_ = S1 + (S2 × t/180 seconds), where S1 is the speed reached in the last completed stage, S2 is the increment in the treadmill speed at each stage, and t is the time spent (in seconds) in the incomplete stage. The external work was calculated in Joules as bm × g × s × sinθ × t, where bm is the animal’s body mass (kg), g is the acceleration of gravity (9.8 m·s^−2^), s is the treadmill speed (m·min^−1^), θ is the angle of treadmill inclination, and t is the time spent in each stage [12,17]. Values for workload were calculated for each stage of incremental exercise, including the incomplete stage, and were then summed; the value obtained after the summation corresponded to the total external work.

The gross oxygen cost of running, which is inversely associated with the running economy, was calculated by dividing the VO_2_ (mL⋅kg^−1^⋅min^−1^) by the running speed (m⋅min^−1^), and the resulting value was expressed in mL⋅kg^−1^⋅m^−1^ [12,18]. Although the oxygen cost of running is commonly determined during constant-speed exercises, we calculated this parameter in the last minute of the 3-min stages during the incremental exercise, since Wisloff et al. [13] reported that VO_2_ leveled off in running rats ~3 min after the external work was changed. Gross mechanical efficiency was calculated by dividing the caloric equivalent of external work by the caloric equivalent of VO_2_ [19]; the value obtained was then multiplied by 100. According to Brooks et al. [19], the gross efficiency of rats running on a treadmill ranges from 1.3% to 4.5%.

### 2.6. Statistical Analysis

The variables studied were tested for normality using the Shapiro–Wilk test. All variables were normally distributed; therefore, the data are expressed as the means ± SEM. The data presented in Table 1 (i.e., nutritional parameters, physical performance indices, VO_2MAX_, oxygen cost of running, and mechanical efficiency) were compared between the two groups using unpaired Student’s *t*-tests. The VO_2_ curves were compared between the two experimental groups and across time points using two-way analysis of variance (ANOVA) with repeated measures for only the time factor. Tukey’s test was used as the post hoc analysis. The significance level was set at *α* < 0.05. The effect sizes (Cohen’s *d* for two independent means) were calculated for the data presented in Table 1. Effect size allowed the assessment of the magnitude of differences between experimental trials and was calculated by subtracting the mean value of one group from the mean value of the group it was being compared to. The result was then divided by a combined standard deviation for the data. The effect size values were classified as trivial (*ES* < 0.2), small (*ES* = 0.2–0.6), moderate (*ES* = 0.6–1.2), or large (*ES* ≥ 1.2) [20].

## 3. Results

### 3.1. Body Mass Gain and Chow Intake

The initial body mass of the rats was similar between groups (NS: 296.7 ± 4.2 g vs. Sb: 297.6 ± 5.4 g; *p* = 0.887). The control, non-supplemented rats had an average chow intake of 28.3 g per day and gained an average of 3.4 g per day. The group supplemented with Sb did not show a change in chow intake or body mass gain during the 10 days of treatment relative to the group treated with saline (Table 1); indeed, the intergroup differences were classified as trivial or small. These data indicate that the animals tolerated the supplementation well and that the experimental findings can be attributed exclusively to the actions of Sb.

### 3.2. Resting Experiments

A significant main effect of time was observed for VO_2_ data in resting rats (F = 9.859; *p* < 0.001). When the rats were placed on the treadmill, they displayed an average VO_2_ that was above 30 mLO_2_·kg^−1^·min^−1^ irrespective of their experimental group. Then, the VO_2_ gradually decreased until values of 26.95 ± 1.55 and 24.13 ± 1.26 mLO_2_·kg^−1^·min^−1^ for the NS and Sb groups, respectively, were reached after the 90-min rest period (Figure 1A). There was no difference in VO_2_ between the two groups throughout the resting period (F = 0.926; *p* = 0.346) and no interaction between time and groups (F = 0.754; *p* = 0.949).

### 3.3. Incremental-Speed Exercise Sessions

The duration of fatiguing and incremental exercise differed between the two groups; the Sb group ran approximately 8 min longer than the NS group (Table 1). Supplementation with Sb also increased the S_MAX_ attained and the external work performed by the rats by 12.4% and 33.6%, respectively. The improvements caused by Sb in the three indices of aerobic performance were classified as moderate.

A significant interaction between time and group was observed for VO_2_ data in rats subjected to exercise (F = 3.725; *p* < 0.001). Treadmill running induced an immediate and marked increase in VO_2_ in both groups. The VO_2_ value was not different between the two groups up to the 27th minute of exercise when all rats from both groups were still running. However, the VO_2_ value attained at fatigue, which corresponded to the VO_2MAX_, was 12.7% higher in the Sb group than in the NS group (68.82 ± 2.08 mLO_2_·kg^−1^·min^−1^ vs. 61.09 ± 1.83 mLO_2_·kg^−1^·min^−1^; *p* = 0.010; Figure 1B); this difference in VO_2MAX_ was also classified as moderate. In agreement with the latter findings, significant and positive correlations were observed between VO_2MAX_ and the three aerobic performance indices evaluated: Time to fatigue (r = 0.778; *p* < 0.001; Figure 1C), S_MAX_ (r = 0.778; *p* < 0.001), and total external work (r = 0.752; *p* < 0.001).

We then calculated the oxygen cost of running (which represents the running economy) and mechanical efficiency in an attempt to better understand our data. Regardless of the timing of the calculation (in the first stage or last completed stage of the exercise), no significant intergroup differences were observed (Table 1; trivial and small-magnitude differences). In addition, we investigated whether these parameters were associated with VO_2MAX_. No significant correlations were observed between VO_2MAX_ and running economy (r = 0.099; *p* = 0.631) or mechanical efficiency (r = −0.051; *p* = 0.804) measured in the last completed stage.

## 4. Discussion

In the present study, supplementation with Sb did not alter the VO_2_ of rats kept undisturbed on the treadmill belt. In addition, during submaximal intensities of the incremental-speed running exercise, the probiotic did not change the VO_2_ values, running economy, or mechanical efficiency. In contrast, at fatigue, VO_2MAX_ was increased in supplemented rats compared with controls, despite the absence of changes in running economy or mechanical efficiency. Interestingly, Sb supplementation improved aerobic performance, as evidenced by the increases in time to fatigue, maximal speed attained, and external work performed by the supplemented rats during the incremental exercise.

The lack of changes in the resting VO_2_ of the supplemented rats suggests that the beneficial metabolic effects mediated by Sb are not due to changes in energy expenditure via oxidative pathways. It is relevant to state that the rats used in the present experiments were young, fed a conventional chow diet, and likely did not exhibit any metabolic dysfunction. This is an important difference from previous studies in which beneficial effects of Sb on metabolism were observed; these studies used hamsters subjected to a cholesterol-enriched diet [4] or obese and diabetic mice [5]. Thus, further investigation into the effects of Sb on the aerobic metabolism of unhealthy animals or human participants is warranted.

The running time to fatigue and VO_2MAX_ were, respectively, 21.6% and 12.7% higher in supplemented rats compared to non-supplemented rats. Because no other study has investigated whether Sb supplementation influences physical performance, the following discussion was based on the effects of different probiotics administered at doses similar to that used in the present study. Our findings are in agreement with those reported by Chen et al. [21]; these authors observed that the endurance swimming time in mice was increased following six weeks of oral administration of *Lactobacilllus plantarum* TWK10 compared with control mice treated with a vehicle. Similarly, in humans, Shing et al. [8] reported an extended time to fatigue in trained runners during a fixed-intensity exercise performed in a hot environment following 4 weeks of supplementation with a multi-strain probiotic (*Lactobacillus, Bifidobacterium*, and *Streptococcus* strains). Nevertheless, an improvement in physical performance is not a universal finding after supplementation with probiotics. Cox et al. [9] did not report any differences in the treadmill performance of trained distance runners following 4 weeks of supplementation with *Lactobacillus fermentum* VRI-003 relative to the placebo treatment. The differences between our findings and those of previously published studies may be explained by methodological differences. Factors, such as probiotic type (bacteria or yeast species), supplementation time and duration, exercise type, species (human, rat, or mouse), and sample characteristics, may influence the experimental outcomes [22].

Shing et al. [8] suggested the maintenance of gastrointestinal structural integrity, endotoxin translocation, and immune modulation as possible causes for the probiotic-mediated improvement of performance in the heat. Indeed, these factors are suggested to modulate performance when animals are subjected to high thermoregulatory strain, particularly during prolonged exercise under uncompensable heat stress. However, because our experiments were conducted in a temperate environment (24 °C) and the exercise consisted of an incremental-speed running exercise, these factors likely do not explain the ergogenic effects mediated by Sb. Under the present conditions, the ability of the cardiovascular system to specifically increase coronary and skeletal muscle blood flow is more of a determinant for physical performance than thermoregulatory responses [16].

A plausible hypothesis to explain the increase in VO_2MAX_ caused by Sb is a greater ability to transform energy using lipid substrates. Sb supplementation increases the fecal concentrations of SCFAs [6], which have several modulatory actions on metabolism. For example, mice fed a high-fat diet supplemented with sodium butyrate, an SCFA, present increased levels of phosphorylated adenosine monophosphate-activated protein kinase (AMPK) in oxidative muscles [23]. The greater activation of AMPK stimulates peroxisome proliferator-activated receptor-gamma coactivator-1α (PGC-1α) [23], which, in turn, promotes the gene expression of proteins involved in lipid oxidation and mitochondrial respiration. The latter responses represent a physiological mechanism that may underlie the increased aerobic performance in Sb-treated rats, although we recognize that rodents fed a standard diet will oxidize carbohydrates to generate energy at high exercise intensities [13], including the intensities close to the maximal aerobic speed.

Although Sb increased VO_2MAX_ in our rats, the running economy and mechanical efficiency remained unaltered. Notably, aerobically trained rats [12] or rats with higher intrinsic exercise capacity [24] perform better and spend less energy while running on a treadmill than their respective controls. Improvements in running economy and mechanical efficiency following aerobic training are explained by biomechanical factors, including improved technique and the transfer of elastic energy during stretch–shortening cycles [25], as well as by physiological adaptations in skeletal muscle, including an increase in mitochondrial content, which results in increased respiratory capacity [26,27]. Thus, Sb supplementation does not seem to affect the biomechanical and physiological aspects mentioned above, and most likely, the Sb-mediated increase in VO_2MAX_ does not result from activation of the AMPK/PGC-1α pathway.

The significant and positive correlation between VO_2MAX_ and time to fatigue does not allow us to identify which one is the cause and which one is the consequence. The rationale presented in the previous paragraphs supports the idea that increased muscle function, characterized by greater VO_2MAX_, leads to improved performance. In contrast, one can argue that increased performance may have resulted from changes in non-metabolic factors (e.g., motivation), forcing rats to reach greater rates of oxygen consumption. This rationale (discussed in the next paragraph) comes with the assumption that non-supplemented rats did not attain their VO_2MAX_ during the incremental exercise. Thus, the greater performance caused by supplementation with probiotics may also have resulted from changes in neural function.

Evidence indicates that the ingestion of probiotics is associated with changes in the concentrations of serotonin and dopamine metabolites in specific brain areas [28] and promotes anxiolytic-like effects in rats and beneficial psychological effects in healthy human volunteers [29]. In fact, there is evidence pointing to a potential impact of the enteric microbiota on brain function [30]; more specifically, many of the gut microbiota or potential probiotics-mediated effects on brain function are dependent on vagal activation [31], although vagus-independent mechanisms have already been reported [32]. Therefore, it is possible that the ergogenic effects mediated by Sb may be dependent on the modulation of serotonergic and dopaminergic neurotransmission, both of which have been associated with aerobic performance [33]. As previously identified, Sb administration changes gut microbiota composition [5,34], which may affect the activity of vagal afferent fibers in the gut, thereby influencing monoaminergic brain systems [35]. This hypothesis is based on similar observations regarding physical performance and VO_2MAX_ of rats treated with central dopamine [36]. If this hypothesis is correct, increased motivation to exercise would allow the rats to achieve faster speeds and therefore, a higher VO_2MAX_.

Another hypothesis that could explain the increased performance in Sb-treated rats is the modulation of immune function, as evidenced by the observations that Sb supplementation reduces the concentrations of proinflammatory cytokines, including interleukin-8 (IL-8), IL-6, IL-1β, tumor necrosis factor alpha (TNF-α), and interferon gamma (IFN-γ) [37,38,39,40]. Recently, a neuroinflammatory model was proposed by Vargas and Marino [41] to explain fatigue during an acute physical exercise session. In this model, augmented concentrations of circulating IL-6 and other inflammatory mediators would contribute to increased perceived fatigue during exercise, thus leading to less muscle recruitment and consequently, reduced aerobic performance.

Of note, all the hypotheses raised to explain the augmented performance and VO_2MAX_ in supplemented rats are merely speculative because we did not analyze the concentrations of intramuscular substrates, cytokines, and neurotransmitters. Future investigations should identify the mechanisms underlying the benefits mediated by Sb and evaluate whether the present findings can be translated to human physiology. At the present moment, our findings cannot be used to endorse Sb supplementation as an efficient strategy for improving aerobic performance of recreational or professional athletes. In addition, it is not clear whether the Sb-induced increases in VO_2MAX_ and S_MAX_ are indeed beneficial outcomes, as these increases were not accompanied by adaptations that are commonly caused by aerobic training, such as greater running economy and mechanical efficiency. For example, a dopamine/noradrenaline reuptake inhibitor enhances aerobic performance but also core body temperatures during exercise in a hot environment, suggesting that this drug overrides the inhibitory signals arising from the central nervous system that cause exercise to stop when close to critical temperature values [42]. This may also be the case for Sb supplementation, which may allow the rats to exercise beyond safe limits.

## 5. Conclusions

We conclude that Sb supplementation does not affect resting aerobic metabolism but does increase VO_2MAX_ and aerobic performance in rats. In addition, the present data rule out the suggestion that increased running economy and mechanical efficiency could explain the enhanced performance of supplemented rats.

## Figures and Tables

**Figure 1 nutrients-11-02352-f001:**
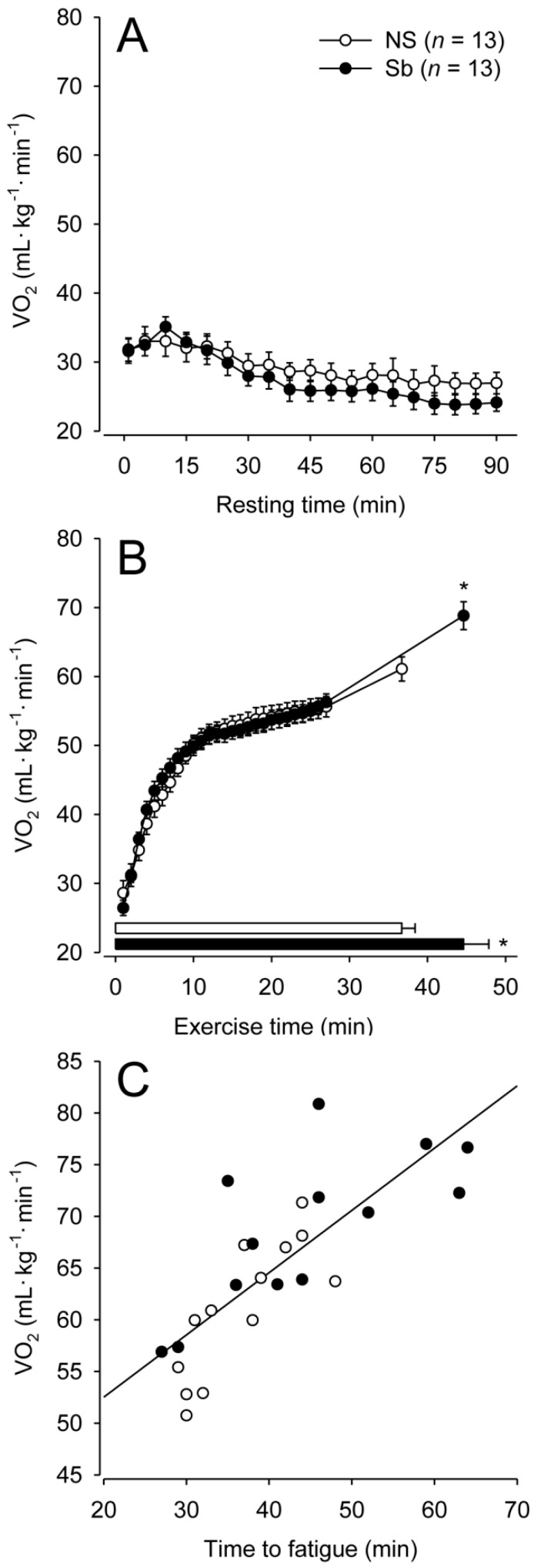
The rate of oxygen consumption in rats during the 90-min rest (Figure 1**A**) or in rats subjected to an incremental-speed exercise (Figure 1**B**). NS, non-supplemented group (○); Sb, *Saccharomyces boulardii*-supplemented group (●). Data are expressed as the means ± SEM. The horizontal bars at the bottom of panel B represent the time to fatigue. * denotes a significant difference compared with the NS group. Figure 1**C** shows the significant and positive correlation between the maximal rate of oxygen consumption (VO_2MAX_) and time to fatigue. Data are expressed as individual values.

**Table 1 nutrients-11-02352-t001:** The nutritional parameters during the 10 days before the experimental trials, the physical performance indices, running economy, and the mechanical efficiency values of rats from the non-supplemented (NS) and the *Saccharomyces boulardii*-supplemented (Sb) groups during the incremental exercise.

Parameter	NS	Sb	*p*-Value	Cohen’s *d*
Nutritional parameters				
Body mass gain (g)	33.6 ± 4.8	34.9 ± 3.7	0.827	0.084
Chow intake (g·day^−1^)	28.3 ± 1.9	26.4 ± 0.5	0.357	0.381
Physical performance indices				
Time to fatigue (min)	36.7 ± 1.8	44.6 ± 3.4	0.047	0.816
Maximum speed attained (m·min^−1^)	21.2 ± 0.6	23.9 ± 1.1	0.047	0.845
Total external work (J)	162.9 ± 10.6	217.6 ± 22.1	0.036	0.874
VO_2MAX_ (mLO_2_·kg^−1^·min^−1^)	61.09 ± 1.83	68.82 ± 2.08	0.010	1.095
Gross oxygen cost of running				
First stage (mLO_2_·kg^−1^·m^−1^)	3.48 ± 0.15	3.64 ± 0.10	0.392	0.349
Last stage (mLO_2_·kg^−1^·m^−1^)	2.91 ± 0.05	2.91 ± 0.08	0.965	0.000
Gross mechanical efficiency				
First stage (%)	1.24 ± 0.07	1.16 ± 0.03	0.315	0.414
Last stage (%)	1.40 ± 0.03	1.40 ± 0.04	0.908	0.000

Data are expressed as the means ± SEM.

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
