# Peer review of "Supplementation with Saccharomyces boulardii Increases the Maximal Oxygen Consumption and Maximal Aerobic Speed Attained by Rats Subjected to an Incremental-Speed Exercise"

_nutrients, 2019, doi:10.3390/nu11102352_

Round 1

Reviewer 1 Report

The present paper reports the effect of a starch-based probiotic and the impact on VO2max and other aerobic factors in rats. This manuscript is very well written with descriptions of the methods in a replicable context. The methods and statistical analyses are justified as well as equations or conversion factors. The methods were appropriately determined to address the primary purpose of this research study and the discussion substantially supports the reported results. That being said, I have a few (minimal) comments and questions that need to be addressed.

Throughout the manuscript, there are several run-on sentences, lacking punctuation or words (was, as, and) and the overly long sentence tends to lose the reader in what the point of the statement is. A careful read-through can alleviate some of the issues and improve readability.

Abstract: Can the authors please add p-values to the abstract? The data are reported to be significant but it is not clear until the manuscript is read. Often researchers are looking at abstracts to gain insight and will then gain access to full-text.

Introduction: lines 47 and 66-The term “metabolic rate” is used often (throughout the manuscript). I would suggest modification of the term to “aerobic consumption” or “aerobic power” and here is why; The term metabolic rate is often referred to in context of kcal of joules (energy) consumption or utilization, rather than oxygen utilization and because the focus of this study was metabolic efficiency based on oxygen consumption (VO2), then “metabolic rate” is an inaccurate description of the variables assessed.

Methods: Clarification-I am assuming all rats underwent familiarization?

Results: (Table 1, Figure 1, and lines 171, 172, and 193)-The “.” In mL·kg-1·min-1 shifted down as a period rather than per-dot.

Results: It is expected that the rats would run further if they ran for a longer duration. This is not surprising or significant. Nor that speed was higher because they all are a function of one-another. This is why the correlations are nearly identical as well. Therefore, in my opinion, the three performance indices do not need to be separately distinguished in the text as Table 1 is redundant information. However, meaningful results would be the running economy or as the authors phrase it “mechanical efficiency” is not different between the treatment groups. I think this concept and lack of difference in running economy is something that should be highlighted more in the discussion and not just glazed over.

Discussion: The discussion provides adequate substantiation and evidence to support all of the research findings (significant and non-significant). I think the authors did a thorough job at presenting how their data compare to other similar studies.  I do believe this should be evaluated further, either for a longer duration or as a proportionate conversion to application in humans. The results and related discussion material are worth exploring further. That being said is Sb supplementation truly beneficial if running economy is not improved even if VO2 is?

Thank you for your contributions to this topic! Well done.

Author Response

Nutrients-580605 – Response to reviewers

Reviewer #1

The present paper reports the effect of a starch-based probiotic and the impact on VO2max and other aerobic factors in rats. This manuscript is very well written with descriptions of the methods in a replicable context. The methods and statistical analyses are justified as well as equations or conversion factors. The methods were appropriately determined to address the primary purpose of this research study and the discussion substantially supports the reported results. That being said, I have a few (minimal) comments and questions that need to be addressed.

We thank the first reviewer for the relevant comments raised during the review process. We took the report of the reviewer very seriously, and our responses to his/her comments are presented below. We believe that our manuscript was considerably improved and hopefully it will be suitable for publication. 

Throughout the manuscript, there are several run-on sentences, lacking punctuation or words (was, as, and) and the overly long sentence tends to lose the reader in what the point of the statement is. A careful read-through can alleviate some of the issues and improve readability.

We have read the entire manuscript very carefully and made several corrections to alleviate some of the issues mentioned by the reviewer 1 and to improve the text’s readability.

Abstract: Can the authors please add p-values to the abstract? The data are reported to be significant but it is not clear until the manuscript is read. Often researchers are looking at abstracts to gain insight and will then gain access to full-text.

Thank you for the very good suggestion. In the revised abstract, we have included p-values for the comparisons between groups regarding the VO2MAX and maximal speed attained by the rats (lines 29-31).

Introduction: lines 47 and 66-The term “metabolic rate” is used often (throughout the manuscript). I would suggest modification of the term to “aerobic consumption” or “aerobic power” and here is why; The term metabolic rate is often referred to in context of kcal of joules (energy) consumption or utilization, rather than oxygen utilization and because the focus of this study was metabolic efficiency based on oxygen consumption (VO2), then “metabolic rate” is an inaccurate description of the variables assessed.

We agreed with the reviewer that the term “metabolic rate” is an inaccurate description of the variables assessed. Therefore, as suggested, we have replaced this term with more precise terms in the two sentences of the introduction (lines 46 and 70-71). We have also replaced the term “metabolic rate” in two sentences of the discussion (lines 251 and 304-305) and in the first sentence of conclusions (line 346).

Methods: Clarification-I am assuming all rats underwent familiarization?

Yes, the reviewer is correct. All of the rats underwent the protocol for familiarization with treadmill running. This information is now more clearly stated in the Materials and Methods of the revised manuscript (lines 104-105).

Results: (Table 1, Figure 1, and lines 171, 172, and 193)-The “.” In mL·kg-1·min-1 shifted down as a period rather than per-dot.

As suggested by the reviewer, we have corrected the dot used in the measuring unit for VO2MAX throughout the manuscript (lines 151-152, 193-194, and 210), including Table 1 and Figure 1.

Results: It is expected that the rats would run further if they ran for a longer duration. This is not surprising or significant. Nor that speed was higher because they all are a function of one-another. This is why the correlations are nearly identical as well. Therefore, in my opinion, the three performance indices do not need to be separately distinguished in the text as Table 1 is redundant information. However, meaningful results would be the running economy or as the authors phrase it “mechanical efficiency” is not different between the treatment groups. I think this concept and lack of difference in running economy is something that should be highlighted more in the discussion and not just glazed over.

We thank the reviewer for the very good (constructive) suggestions. In order to address these points and the issues raised by the second reviewer, we have replaced the “distance travelled” data with “external work” data in Table 1 and in the Results section (line 201-203). Hopefully this replacement will reduce redundant information in our manuscript, because the calculation of external work takes not only the distance travelled by the rats into account, but also their body masses. Moreover, we have highlighted the lack of difference in running economy in our revised discussion (lines 290-299).

Discussion: The discussion provides adequate substantiation and evidence to support all of the research findings (significant and non-significant). I think the authors did a thorough job at presenting how their data compare to other similar studies. I do believe this should be evaluated further, either for a longer duration or as a proportionate conversion to application in humans. The results and related discussion material are worth exploring further. That being said is Sb supplementation truly beneficial if running economy is not improved even if VO2 is?

Thank you for the nice words regarding our discussion. We have included three sentences in the last paragraph of discussion (lines 337 to 344) to address whether Sb supplementation is truly beneficial.

Reviewer 2 Report

Your paper regarding the use of Sb supplemented to a group of rats vs a control, is interesting and nicely written. I believe you clearly stated that any mechanism discussed related to how Sb might have influenced greater VO2max, max speed, time to fatigue, and distance traveled (vs control) is speculative. Also, it seems your research was adequately controlled - with all being equal except that one group ingested Sb. 

A few, common things or themes I mention in the attached document, where I have comments related to your entire manuscript (per line), are (a) please better define mechanical efficiency (ME) and why not term the "oxygen cost of running" as running economy (RE)? I know your collaborative research group has published other papers related to ME where you nicely define workload and VO2. Further, when you discuss the oxygen cost of movement, relate this to ME and RE. Lastly, you might consider expanding your definition of ME by relating it to the efficiency of muscular work, such that high ME = greater force generation in the locomotor musculature. I realize you did not find statistical difference in ME between groups; thus your speculative discussion is appropriate and helps with future research ideas. 

Please look over the attached document for other changes and clarifications within your manuscript. 

Author Response

Nutrients-580605 – Response to reviewers

Reviewer #2

Comments and Suggestions for Authors

Your paper regarding the use of Sb supplemented to a group of rats vs a control, is interesting and nicely written. I believe you clearly stated that any mechanism discussed related to how Sb might have influenced greater VO2max, max speed, time to fatigue, and distance traveled (vs control) is speculative. Also, it seems your research was adequately controlled - with all being equal except that one group ingested Sb.

We thank the second reviewer for the relevant comments raised during the review process. We took the report of the reviewer very seriously, and our responses to his/her comments are presented below. We believe that our manuscript was considerably improved and hopefully it will be suitable for publication. 

A few, common things or themes I mention in the attached document, where I have comments related to your entire manuscript (per line), are (a) please better define mechanical efficiency (ME) and why not term the "oxygen cost of running" as running economy (RE)? I know your collaborative research group has published other papers related to ME where you nicely define workload and VO2. Further, when you discuss the oxygen cost of movement, relate this to ME and RE. Lastly, you might consider expanding your definition of ME by relating it to the efficiency of muscular work, such that high ME = greater force generation in the locomotor musculature. I realize you did not find statistical difference in ME between groups; thus your speculative discussion is appropriate and helps with future research ideas.

Thank you for the careful revision of the manuscript. Our responses to all of the comments are presented below in a point-by-point fashion. In general lines, it is more clearly stated in the revised manuscript that the terms “oxygen cost of running” and “running economy” are basically synonyms. In addition, we calculated the gross mechanical efficiency and inserted these data in the manuscript.

Line 21 – your use of “mechanical efficiency” at this point in the abstract is vague; your collaborative/research group seems to have published several articles related to ME. I suggest defining ME right away for the reader, even in the abstract, then later in methods using the exact formula used (such as: ME (%) = (workload / VË™ O2) ·100; with a complete equation for the determination of workload. But, at least in the abstract, for instance, say ME (%) = (workload / VË™ O2) ·100.

As suggested by the reviewer, the abstract now contains the equation used to calculate mechanical efficiency (lines 21-22). We used the equation proposed by Brooks et al. [J Appl Physiol Respir Environ Exerc Physiol. 1984, 56, 520-525]. This is the seminal study that has calculated gross mechanical efficiency in rats subjected to treadmill running. Meanwhile, we have included the equations used to calculate both the external work and mechanical efficiency in the Materials and Methods section of the revised manuscript (lines 144-149 and 155-158).

Line 37 – delete space between reference numbers (I think this is the correct format; check to be sure).

Thank you for the careful revision of our manuscript. We have deleted the space between reference numbers throughout the revised manuscript.

Line 58 – You need a reference for this sentence and please describe in more detail what you mean by “better muscle function”. It’s good you give an example in the next sentence, but it’s important to explain “better muscle function” in a broader sense before giving a specific example. Also, the mechanisms or variables underlying a better aerobic performance are numerous and you only state “muscle function”. Consider reminding the reader of a few more, primary aerobic performance indicators (e.g., high VO2max, enhanced running economy, high lactate threshold – the “3” key parameters of any Performance Model for endurance athletes; with rat research, it’s important to attempt to relate your findings to humans, which you do, briefly, at the end.)

We agreed with the reviewer that the expression “better muscle function” lacks specificity / clarity. Therefore, this expression was deleted, and the fourth paragraph of the introduction was basically rewritten in the revised manuscript (lines 58-69). This paragraph now describes some skeletal muscle adaptations induced by aerobic training, as well as the key determinant parameters that regulate aerobic performance.

Line 63 – Again, I think the correct format is no space between reference numbers. Check this to be sure and implement throughout your manuscript as necessary.

Again, thank you for the careful revision of our manuscript. We have deleted the space between reference numbers throughout the revised manuscript. Moreover, all the references cited in the text were double-checked for correctness.

Line 70 – when you say “lower oxygen cost of running” you mean better or enhanced running economy (RE), please indicate as such. ME is different than running economy, but high ME should equal high RE. Try to make this clear in your manuscript.

The sentence was amended (lines 72-73), as requested by the reviewer.

Line 165 – you need to define and actually calculate mechanical efficiency (ME) in your table, such that ME (%) = (workload / VË™ O2) ·100 [per other research from your collaborative group, such as: (a) Rabelo PCR, Cordeiro LMS, Aquino NSS, Fonseca BBB, Coimbra CC, Wanner SP, Szawka RE, Soares DD. Rats with higher intrinsic exercise capacities exhibit greater preoptic dopamine levels and greater mechanical and thermoregulatory efficiencies while running. J Appl Physiol (1985). 2019 Feb 1;126(2):393-402. doi: 10.1152/japplphysiol.00092.2018. Epub 2018 Jun 21; or (b) Soares DD, Lima NR, Coimbra CC, Marubayashi U. Evidence that tryptophan reduces mechanical efficiency and running performance in rats. Pharmacol Biochem Behav 74: 357–362, 2003. doi:10.1016/S0091-3057(02)01003-1.]. Overall, it appears workload from article (a) is best, and was calculated: workload (Joule) = body wt·g·s·sinθ·t, where: body wt (kg), g = gravity force (9.8 m/s2), s = speed (m/min), θ = inclination of the treadmill (5°), t = time spent in each stage in minute.

Thank you for the very good suggestion. We have included the equations used to calculate both the external work and gross mechanical efficiency in the Materials and Methods section of the revised manuscript (lines 144-149 and 155-158). To calculate external work (or workload), we used the equation cited by Rabelo et al. (J Appl Physiol. 2019, 126, 393-402). Whereas to calculate gross mechanical efficiency, we used the equation proposed by Brooks et al. [J Appl Physiol Respir Environ Exerc Physiol. 1984, 56, 520-525]. This is the seminal study that has calculated gross mechanical efficiency in rats subjected to treadmill running.

Moreover, both the data regarding running economy and mechanical efficiency were included in Table 1 and are now described in the Results section (lines 229-235).

Line 198 – after reading through your results (nice work), in this line you refer to the oxygen cost of running, which is running economy (RE) (different than mechanical efficiency, which refers to, as you’re aware, ME = workload / VO2. Clarify this.

The results section was amended to improve clarity (lines 229-235).

Line 205 – here again, you mention “oxygen cost of running” when you need to relate it to running economy.

As suggested, this sentence was amended.

Line 216 – by subjects, do you mean “human participants”? Be specific and state human participants. Delete subjects.

Thank you for the suggestion. We have replaced “subjects” with “human participants” (line 252).

Line 220 – You say “…Lactobacilllus plantarum TWK10”, which was from another study. Please specify if this was in the same dose per kilogram of body weight as your study. Dose of supplement should matter, and especially expressing it per kilogram of body weight gives the reader a sense of “dose” and if the other study is, indeed, showing the same response as your study, where all things should be equal. Also, state whether the other study had a control group like your methodology.

The dose of Sb used in the present study corresponded to approximately 3 ´ 108 CFU·kg-1·day-1, which is similar to the lower dose of probiotic (2.05 ´ 108 CFU·kg-1·day-1) used in the study of Chen et al. (Nutrients. 2016, 8, 205). This information was included in the revised discussion (lines 254-257). In addition, we are now stating that the study of Chen et al. had a control group (lines 257-259).

Lines 227-230 – need a reference to back your claims up.

As requested, we inserted a reference to support our claims (line 269).

Lines 231-232 – here you begin to broaden the discussion to probiotics in general with the use of a reference; how does your current supplement (Sb) compare to probiotics in general? Please explain, in your paper, how discussing different supplements (i.e., different than Sb) and/or generalized probiotic intake is relevant to your, individual supplement (i.e., Sb)? Most credibly, you should refrain from discussing different doses and different supplements other than Sb-oriented research; unless you can somehow standardize the quantification and quality of Sb vs other general probiotics?

We agreed with the point raised by the reviewer. However, we could not find other studies that have investigated whether Sb supplementation influences physical performance. Therefore, our discussion was based on the effects of different probiotics administered at doses similar to the doses used in the present study; the doses investigated ranged from 2.05 ´ 108 to 1.03 ´ 109 CFU·kg-1·day-1 in mice and from 1.2 to 4.5 x 1010 CFU·kg-1·day-1 in humans (of note, the comparison of doses used for rats and human participants should consider the conversion from a human equivalent dose). We are now emphasizing that the dose used by Chen et al. was similar to the dose used in the present study (lines 254-257).

We respectfully ask the reviewer to allow us to maintain this paragraph, because it is relevant to state that modulation of microbiota with different probiotics may interfere with physical (aerobic) performance.

Lines 240-250 – in this paragraph you discuss the use of SCFAs at high intensity (to max) exercise. It would be interesting and informative to the reader to also discuss how you reconcile the fact that high carbohydrate use (at least in trained humans on a normal 50-60% Carb, 15% protein, and ~30% fat diet) is favored at ever increasing intensities; in other words, in humans, only individuals habituated to a high fat or keto-type diet will favor fat at high exercise intensities (e.g., > 85% of VO2max). What happens in rats on a high fat diet? Is it the same as in humans? Please clarify.

This is another very good point raised by the reviewer. Therefore, we are stating in the discussion that the hypothesis relating the improved performance to a greater ability to transform energy using lipid substrates seems to be unlikely, because rodents fed a standard diet will oxidize carbohydrates to generate energy at high exercise intensities, including the intensities close to the maximal aerobic speed (lines 286-289). A previous study has reported that trained or untrained rats subjected to a high-fat diet containing a moderate amount of carbohydrates do not show enhanced endurance (Helge et al. J Appl Physiol, 1998, 85, 1342-1348). In addition, these authors also reported the existence of a shift toward increased fat utilization during exercise in the high-fat-fed sedentary or trained rats, being the magnitude of this shift dependent on the type of dietary fat.

Line 243 – delete “s” on SCFAs.

As suggested, the “s” was deleted in the abbreviation (line 282)

Lines 248-249 – again you mention the oxygen cost of running; why not term this running economy, which is related to mechanical efficiency?

As suggested, this sentence was amended (lines 290-291).

Line 263 – you mention the entire microbiota on brain function…in particular, how is this related to Sb? Since you singled out only Sb in your study, it seems like it would be difficult to speculate that Sb could have an effect on brain function and/or motivation because of the complex make-up of the “microbiota”. Please make clear to the reader how Sb alone might have an effect on brain function, among other things.

There is evidence showing that Sb administration changes gut enteric microbiota composition, which may have a potential impact on brain function. More specifically, many of the gut microbiota or probiotics-mediated effects on brain function are dependent on vagal activation, although vagus-independent mechanisms have already been reported. For example, changes in the enteric microbiota composition could affect afferent signaling arising from the vagus nerve and directed to the central nervous system, thereby modulating the serotonergic and dopaminergic neurotransmissions, both of which have been associated with aerobic performance.

The references for the above-information are cited in the revised manuscript. To address the comment raised by the reviewer, we have modified the eight paragraph of the discussion (lines 319-322).

Lines 272-279 – good paragraph. This is where you related an Sb-specific research study to the modulation of the immune system. Thus, your discussion here is stronger, versus relating Sb to generalized probiotic studies. Therefore, use this as a handrail to modify other paragraphs where you talk about generalized probiotics. Try to keep your discussion and conclusion focused on Sb-related studies. Additionally, you could attempt to briefly clarify how discussing broad probiotic studies might relate to Sb, per your study group’ expertise in this area (per other published, collaborative papers).

Thank you for the nice words regarding our paragraph. We agree with reviewer’s concern. However, we emphasize that we could not find other studies that have investigated whether Sb supplementation influences physical performance. In this context, we kindly ask the reviewer to allow us to maintain the third paragraph of the discussion, because it is relevant to state that modulation of microbiota with different probiotics (even generalized probiotics) may interfere with aerobic performance.

Lines 272-279 – good paragraph. This is where you related an Sb-specific research study to the modulation of the immune system. Thus, your discussion here is stronger, versus relating Sb to generalized probiotic studies. Therefore, use this as a handrail to modify other paragraphs where you talk about generalized probiotics. Try to keep your discussion and conclusion focused on Sb-related studies. Additionally, you could attempt to briefly clarify how discussing broad probiotic studies might relate to Sb, per your study group’ expertise in this area (per other published, collaborative papers).

Thank you for the nice words regarding our paragraph. We agree with reviewer’s concern. However, we emphasize that we could not find other studies that have investigated whether Sb supplementation influences physical performance. In this context, we kindly ask the reviewer to allow us to maintain the third paragraph of the discussion, because it is relevant to state that modulation of microbiota with different probiotics (even generalized probiotics) may interfere with aerobic performance.
